# Predictive Factors of Renal Function Decline in Patients with Type 2 Diabetes Treated with Canagliflozin in the Real-Wecan Study

**DOI:** 10.3390/jcm11195622

**Published:** 2022-09-24

**Authors:** Juan J. Gorgojo-Martinez, Miguel Brito-Sanfiel, Teresa Antón-Bravo, Alba Galdón Sanz-Pastor, Jaime Wong-Cruz, Manuel A. Gargallo Fernández

**Affiliations:** 1Department of Endocrinology and Nutrition, Hospital Universitario Fundación Alcorcón, Alcorcón, 28922 Madrid, Spain; 2Department of Endocrinology and Nutrition, Hospital Universitario Puerta de Hierro, Majadahonda, 28222 Madrid, Spain; 3Department of Endocrinology and Nutrition, Hospital Universitario de Móstoles, Móstoles, 28935 Madrid, Spain; 4Department of Endocrinology and Nutrition, Fundación Jiménez Díaz, 28040 Madrid, Spain; 5Department of Endocrinology and Nutrition, Hospital Universitario Infanta Leonor, 28031 Madrid, Spain

**Keywords:** SGLT-2 inhibitor, canagliflozin, type 2 diabetes, renal function

## Abstract

The Real-WECAN study evaluated the real-life effectiveness and safety of canagliflozin 100 mg daily (initiated in SGLT-2 inhibitors naïve patients) and canagliflozin 300 mg daily (switching from canagliflozin 100 mg or other SGLT-2 inhibitors) in individuals with type 2 diabetes. The objectives of this sub-analysis were to estimate the eGFR slope over the follow-up period and to identify predictive factors of eGFR decline in a multiple linear regression analysis. A total of 583 patients (279 on canagliflozin 100 mg and 304 on canagliflozin 300 mg) were included, with median follow-up at 13 months. The patients had a mean age of 60.4 years, HbA1c of 7.76%, BMI of 34.7 kg/m^2^, eGFR below 60 mL/min/1.73 m^2^ 8.6%, and urine albumin-to-creatinine ratio (UACR) above 30 mg/g 22.8%. eGFR decreased by −1.9 mL/min/1.73 m^2^ (*p* < 0.0001) by the end of the study. The mean eGFR slope during the maintenance phase was −0.16 mL/min/1.73 m^2^ per year. There were no significant differences between both doses of canagliflozin in the eGFR reduction or in the eGFR slope. The best predictive multivariate model of eGFR decline after canagliflozin therapy included age, hypertension, combined hyperlipidemia, heart failure, eGFR and severely increased albuminuria. All these variables except hypertension were independently associated with the outcome. In conclusion, in this real-world study, individuals with older age, combined hyperlipidemia, heart failure, higher eGFR and UACR > 300 mg/g showed a greater decline in their eGFR after canagliflozin treatment.

## 1. Introduction

Chronic kidney disease (CKD) in patients with type 2 diabetes (T2DM) not only increases the risk of developing end-stage kidney disease, but also the risk of cardiovascular morbidity and mortality in comparison to people with T2DM but without CKD [1]. Several large cardiovascular and renal outcome trials with sodium–glucose co-transporter type 2 inhibitors (SGLT-2is) in patients with T2DM have shown a significant improvement in composite kidney outcomes and a smaller decline of estimated glomerular filtration rate (eGFR) in comparison to placebo [2]. Proposed mechanisms responsible for renal protection with SGLT-2is are probably multifactorial and include reductions in blood pressure (BP), body weight and HbA1c, as well as a decrease in intraglomerular pressure due to tubuleglomerular feedback activation, a reduction in albuminuria, improved cardiac function with maintenance of renal perfusion, activation of anti-inflammatory and antifibrotic pathways and better oxygenation of tubular cells in the proximal tubule [3].

SGLT-2is induce a reversible acute drop in eGFR, as a result of a reduction in intraglomerular pressure, followed by an attenuation in eGFR decline on long-term treatment. Nevertheless, a high number of patients on SGLT-2is maintain a substantial renal residual risk and their eGFR continues to drop at a rapid rate. Recently, a sub-analysis of the CREDENCE study showed that participants with an acute drop in eGFR after canagliflozin 100 mg were older, had a longer duration of T2DM, had a higher body mass index, systolic BP and eGFR, were more likely to take diuretics and were less likely to have heart failure [4]. However, baseline predictors of long-term eGFR trajectories in patients on SGLT-2is are largely unknown.

In the multicentric observational study Real-WECAN, canagliflozin 100 mg/d (as add-on therapy) and canagliflozin 300 mg/d (switching from canagliflozin 100 mg or other SGLT-2is) were associated with significant improvements in glycemic control, body weight, BP, lipid profile and hepatic biomarkers in patients with T2DM. A significant decrease in the urinary albumin/creatinine ratio (UACR) was found with canagliflozin 100 mg and 300 mg in the subgroup of patients with UACR higher than 30 mg/g Cr, even though there were no meaningful changes in antihypertensive medications during the study [5].

We conducted a post hoc analysis of the Real-WECAN study in order to determine baseline predictors of renal function decline over the follow-up period in patients with T2DM treated with canagliflozin.

## 2. Material and Methods

### 2.1. Study Design and Patient Population

The study design has previously been described [5]. We carried out a retrospective study of two groups of adult patients with T2DM from 5 tertiary hospitals in Madrid, Spain. A first cohort of patients who started canagliflozin 100 mg and a second cohort of individuals with prior SGLT-2i treatment who switched to canagliflozin 300 mg were identified in medical records from the Departments of Endocrinology between May 2015 and July 2019. The second cohort also included those patients from the first cohort who switched to canagliflozin 300 mg, in order to study the subgroup of individuals with the sequential therapy canagliflozin 100 mg–canagliflozin 300 mg [5]. Both cohorts were merged for the present renal sub-analysis, although a separate statistical study for each individual cohort was also performed.

### 2.2. Outcomes and Study Measures

Three data capture visits were defined over the follow-up period: V1, baseline (canagliflozin 100 mg) or switch (canagliflozin 300 mg); V2, 6 ± 2 months after the start of canagliflozin 100 mg or after switching to canagliflozin 300 mg; and V3, last visit. Baseline clinical parameters included gender, age, duration of the disease, weight, BP, chronic diabetic complications, other cardiovascular risk factors, heart failure, sleep apnea, background glucose lowering drugs (GLDs), anti-hypertensive drugs and lipid-lowering medications. Baseline laboratory data included HbA1c, fasting plasma glucose (FPG), heart rate (HR), lipids, uric acid, liver enzymes, hematocrit, eGFR (calculated with the Chronic Kidney Disease Epidemiology Collaboration equation) and UACR. 

The main outcome measures of this sub-analysis were: (1) to estimate the eGFR slope, in order to compare our results with those from other published studies with SGLT-2is, and (2) to determine the best predictive model of eGFR decline at the end of the study, aiming to identify those individuals with high renal residual risk despite canagliflozin therapy.

### 2.3. Statistical Methods

The total eGFR slope was calculated as the annual rate of change in eGFR based on all on-treatment eGFR measurements from baseline to the last available measurement. In order to exclude the acute eGFR drop caused by the hemodynamic effect of canagliflozin, we also calculated the eGFR slope during the maintenance phase as the annual ratio of change in eGFR from visit V2 (6 months) to the end of the study (V3).

A multiple linear regression analysis was used to estimate the best predictive model of eGFR drop over the follow-up period. Forty-seven potential baseline predictors were evaluated. The selection of the best regression equation from all possible sub-models was performed using the Mallows criterion.

## 3. Results

Baseline characteristics are shown in Table 1. A total of 583 patients were included, 279 with canagliflozin 100 mg and 304 with canagliflozin 300 mg; 8.6% of patients had eGFR lower than 60 mL/min/1.73 m^2^ and 22.8% showed UACR higher than 30 mg/g. Median follow-up periods in the canagliflozin 100 mg and canagliflozin 300 mg cohorts were 9.1 and 15.4 months, respectively. Patients switching to canagliflozin 300 mg had been previously treated with other SGLT-2is for a median period of 17.1 months.

There was a modest mean decrease in eGFR (−1.89 mL/min/1.73 m^2^, *p* < 0.0001) at the end of the follow-up period in the entire cohort, which represents a percentage change of −1.67% (Figure 1). This significant reduction in eGFR was observed with both doses of canagliflozin (Figure 2). The mean total eGFR slope was −1.50 mL/min/1.73 m^2^ per year (−1.82 mL/min/1.73 m^2^ per year with canagliflozin 100 mg and −1.27 mL/min/1.73 m^2^ per year after switching to canagliflozin 300 mg). The mean eGFR slope during the maintenance phase was −0.16 mL/min/1.73 m^2^ per year. There were no significant differences between the two doses of canagliflozin in the eGFR reduction or in the eGFR slope. In patients with a follow-up time longer than 2 years (n 139, median time 33.4 months) the percentage change of eGFR was −2.38% and the mean total eGFR slope was −1.0 mL/min/1.73 m^2^ per year. Fifty percent of patients showed no change or increased their eGFR over the follow-up. In the subgroup of patients with baseline eGFR ≥60 mL/min/1.73 m^2^ there was a significant drop in eGFR at the end of the study, whereas a significant rise in renal function was observed in those patients with baseline eGFR below 60 mL/min/1.73 m^2^ (Figure 3).

Fourteen (2.4%) patients (five treated with the 100 mg dose and nine with the 300 mg dose) showed an eGFR decrease greater than 30% at the end of the follow-up period. Baseline characteristics of these patients were similar to those of the entire cohort, except they had a lower baseline eGFR (65.2 vs. 86.5 mL/min/1.73 m^2^, *p* < 0.0001). SGLT-i therapy was identified as the most likely cause of rapid decline of eGFR in this group. Doubling of serum creatinine was found in three patients. Only one patient on canagliflozin 100 mg and one patient on canagliflozin 300 mg stopped the drug due to worsening of renal function. No meaningful changes in any anti-hypertensive medications were reported during the study.

The best predictive model of eGFR decline after canagliflozin therapy included age, hypertension, combined hyperlipidemia, heart failure, eGFR and severely increased albuminuria (Table 2). All these variables except hypertension were independently associated with the outcome. Canagliflozin dose, diuretic therapy or advanced stages of CKD were not associated with a larger decrease in eGFR.

## 4. Discussion

In the present analysis of the Real-WECAN study, renal function slightly lowered over the follow-up period in the entire cohort, but the eGFR slope leveled off over the maintenance phase. These findings are similar to the renal effects reported with different SGLT-2is in several cardiovascular outcome trials, in which most patients, as in our study, had well-preserved kidney function [6,7,8]. For example, in the CANVAS trial, [7] mean change in eGFR in patients who were treated with canagliflozin was −1.8 mL/min/1.73 m^2^ (mean difference 2.0 mL/min/1.73 m^2^ vs. placebo). From baseline to week 13, the canagliflozin-treated group had a mean eGFR decrease of −3.1 mL/min/1·73 m^2^; from week 13 to last available measurement (median 20.9 months), participants allocated to the canagliflozin group had a stabilization of kidney function, with a mean annual long-term change of +0.3 mL/min/1.73 m^2^. In the EMPA-REG OUTCOME trial, [6] there was a short-term drop in the eGFR in the empagliflozin groups (weekly reductions of −0.62 mL/min/1.73 m^2^ in the 10 mg cohort and −0.82 mL/min/1.73 m^2^ in the 25 mg cohort). Thereafter, during long-term administration (median follow-up time 3.1 years), the eGFR remained stable in the empagliflozin groups, with annual decreases of −0.19 mL/min/1.73 m^2^ in the 10 mg and 25 mg empagliflozin groups, as compared to a reduction of −1.67 mL/min/1.73 m^2^ in the placebo group.

The long-term protective effects of SGLT-2is on renal function have also been found in some, but not all, large real-world studies (RWS) [9,10,11]. In the CVD-REAL3, [9] a multinational observational study in which new users of SGLT2is and other GLDs were compared after a propensity score matching, annual rates of eGFR change during a 14.9-month follow-up were +0.46 mL/min/1.73 m^2^ per year in the group of SGLT-2is and −1.21 mL/min/1.73 m^2^ per year in the group of other GLDs. However, in the DARWIN-T2D, [10] a multicenter retrospective study conducted in Italy which compared patients treated with dapagliflozin versus other GLDs in real life, eGFR decreased −1.1 mL/min/1.73 m^2^ in the dapagliflozin cohort vs. −0.6 mL/min/1.73 m^2^ in the other GLDs cohort after an average follow-up of 6 months. The change in eGFR from baseline between both groups was not significantly different. Likewise, an RWS conducted by the Scottish Diabetes Research Network Epidemiology Group [11] showed that, in patients exposed to dapagliflozin (median follow-up 210 days), eGFR dropped by −1.81 mL/min/1.73 m^2^ at 3 months but by 12 months the effect was not higher than the expected reduction in eGFR in the absence of the SGLT-2i. These discrepancies in renal findings among the aforementioned RWS could be explained by differences in the follow-up periods, which were shorter in the DARWIN-T2D and the Scottish study in comparison to the CVD-REAL3 or the REAL-WECAN. A longer observational time is needed to distinguish between the acute versus chronic effects of SGLT-2i on eGFR; most patients in RWS do not have impaired renal function and therefore the rate of deterioration is slow, meaning that it would take a long time to observe the renal benefits of SGLT-2is.

In the REAL-WECAN study, older age, combined hyperlipidemia, heart failure, higher baseline eGFR and severely increased albuminuria were independently associated with a greater long-term decrease in eGFR after canagliflozin therapy. Interestingly, the subgroup of patients with baseline eGFR below 60 mL/min/1.73 m^2^ showed a significant increase in eGFR over the follow-up period. In dedicated renal outcome trials enrolling patients with CKD, such as CREDENCE [12] or DAPA-CKD [13], the change in the estimated GFR slope was less in individuals on SGLT-2i than in those on placebo, but a sustained decline in eGFR over the follow-up period was still observed despite SGLT-2i therapy. Such decline was also present in those patients on SGLT-2is with eGFR below 60 mL/min/1.73 m^2^ [14]. However, some RWS including patients with CKD have shown a long-term improvement in renal function on SGLT-2i therapy. For example, in the SAPPHIRE study, [15] a long-term, post-marketing surveillance conducted in Japan, patients with CKD and G3-G4 stages treated with canagliflozin experienced a slight increase (+0.5–1.1 mL/min/1.73 m^2^) in eGFR after 36 months of treatment, as opposed to patients with G1 stages, who experienced a decrease in eGFR (−6.9 mL/min/1.73 m^2^). In the CVD-REAL3 study, [9] patients with baseline eGFR lower than 60 mL/min/1.73 m^2^ had an eGFR slope of +0.22 mL/min/1.73 m^2^ per year; in contrast, those patients with eGFR > 90 mL/min/1.73 m^2^ showed an eGFR slope of −0.18 mL/min/1.73 m^2^ per year. The mechanisms responsible for this recovery of eGFR in some RWS are unknown. It could be hypothesized that compensatory upregulation of tubular SGLT-1 in advanced stages of CKD may result in increased sodium–glucose co-transport, causing afferent arteriolar re-dilatation and an increase in renal perfusion and eGFR. Alternatively, anti-inflammatory and antifibrotic pathways might be involved in the long-term renal function recovery in this subgroup of patients.

It is well known that the risk of developing end-stage kidney disease is further increased at higher levels of albuminuria [1]. In several clinical trials, the absolute effect of SGLT-2is on eGFR slope was greater in people with severely increased albuminuria and this was due to the much more rapid decline in kidney function in these patients [16,17]. In our study, individuals with UACR >300 mg/g at baseline showed a higher decline in the eGFR in comparison to patients with albuminuria below 30 mg/g, confirming that this group maintains a high renal residual risk despite SGLT-2is therapy.

Heart failure and CKD frequently co-exist because both diseases share common risk factors and pathophysiological mechanisms which affect one another. Heart failure has been associated, after adjustment for other well-known CKD risk factors, with a more pronounced drop in eGFR over time [18]. Although in the CREDENCE trial participants in the canagliflozin group with an acute drop in eGFR were less likely to have heart failure, [4]) our results suggest that, in the long term, these patients remain at a very high risk of developing or worsening CKD.

Metabolic syndrome and its components are associated with CKD [19]. In our cohort, patients with combined hyperlipidemia showed a significant faster decline in their renal function. Hypertension was a significant predictor of greater renal decline in the univariate but not in the multivariate analysis.

As we pointed out in the introduction, several mechanisms may contribute to the nephroprotective effects seen with canagliflozin [3,20]. Well-known metabolic effects include reduction in glucotoxicity and perivisceral fat. Moreover, an increase in beta-hydroxybutyrate levels, a substrate with high energy efficiency for the distal nephron, may protect the renal medulla from hypoxia. Canagliflozin has systemic hemodynamic effects, as it lowers the systolic BP transmitted to the kidney, improves heart function and reduces the need for loop diuretics, which avoids intravascular volume depletion. Intrarenal hemodynamic effects are probably the most important renoprotective mechanism of canagliflozin. Restoration of tubuleglomerular balance, providing more sodium to the macula densa, normalizes the tone of the afferent arteriole and is followed by long-term eGFR stabilization and reduction in albuminuria. Finally, reduction in glucotoxicity and proteinuria and the activation of anti-inflammatory and antifibrotic pathways by canagliflozin may play a joint role in reducing tubulointerstitial damage in the diabetic kidney [20].

We acknowledge the limitations of our analysis. A pre-post design was selected and a control group was not included. However, our main goal was not to compare the effect of canagliflozin with other GLDs, but to identify those patients with high renal residual risk on canagliflozin. Renal function was not available at 3–4 weeks after initiating canagliflozin in most patients, therefore an analysis of the acute eGFR drop could not be performed. Our study focused on patients who started canagliflozin 100 mg and patients with prior SGLT-2i treatment who switched to canagliflozin 300 mg. Unfortunately, eGFR trajectories with other SGLT-2is prior to switching to canagliflozin 300 mg were not collected, so renal data with empagliflozin or dapagliflozin cannot be shown in this analysis. Finally, we cannot rule out the possibility of residual unmeasured confounding factors, despite multivariate adjustments. These limitations may affect the external validity of the results so confirmatory studies including a higher number of patients are needed.

In summary, in the Real-WECAN study, patients with older age, combined hyperlipidemia, heart failure, higher baseline eGFR and severely increased albuminuria showed greater reductions in eGFR after canagliflozin therapy. Our study may help to identify the profile of patients with a higher risk of decline in their renal function after initiating an SGLT-2i. These individuals will need more aggressive control of classic renal risk factors and the combination of SGLT2-is with new nephroprotective drugs.

## Figures and Tables

**Figure 1 jcm-11-05622-f001:**
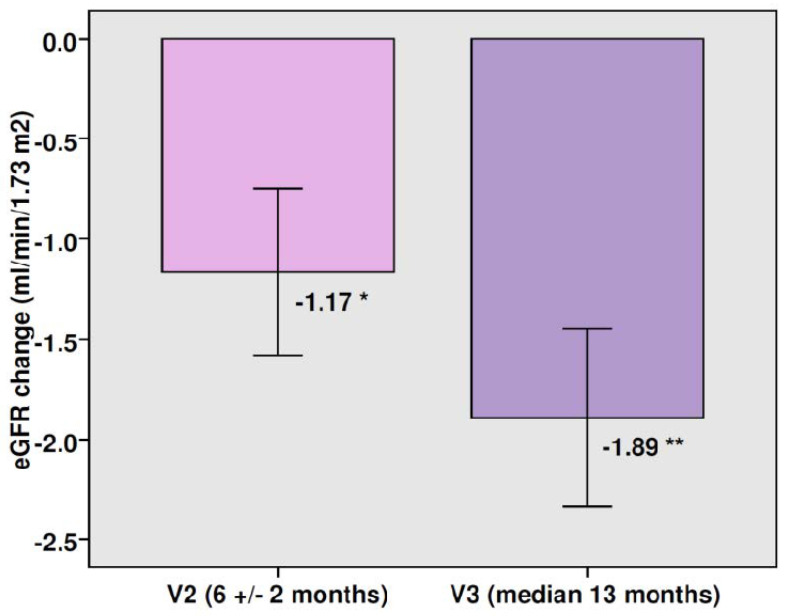
Changes in eGFR after therapy with canagliflozin. Bars: mean change ± SEM. * *p* = 0.005 vs. baseline, ** *p* < 0.0001 vs. baseline.

**Figure 2 jcm-11-05622-f002:**
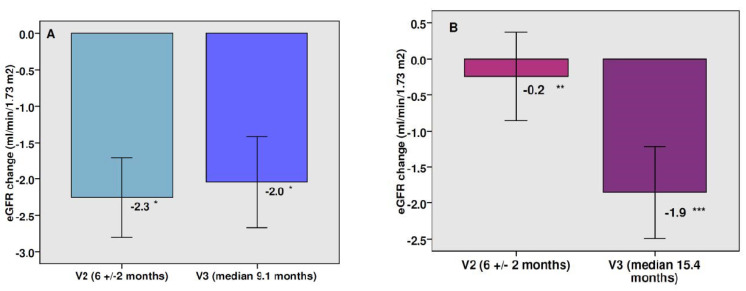
Changes in eGFR after therapy with canagliflozin 100 (panel (**A**)) and after switching to canagliflozin 300 (panel (**B**)). Bars: mean change ± SEM. * *p* < 0.0001 vs. baseline, ** *p* non-significant vs. baseline, *** *p* = 0.001 vs. baseline.

**Figure 3 jcm-11-05622-f003:**
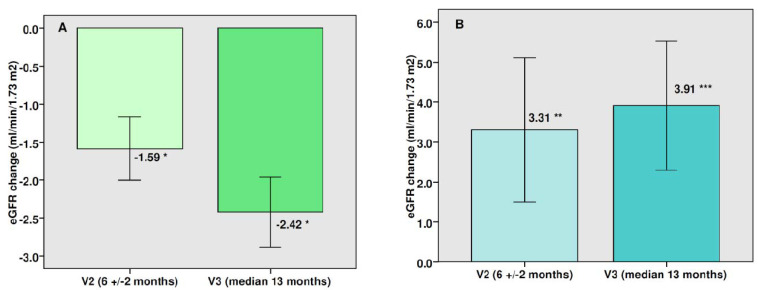
Changes in eGFR after therapy with canagliflozin in patients with baseline eGFR equal to or above 60 mL/min/1.73 m^2^ (91.4%, mean baseline eGFR 88 mL/min/1.73 m^2^, panel (**A**)) and in patients with baseline eGFR below 60 mL/min/1.73 m^2^ (8.6%, mean baseline eGFR 53.5 mL/min/1.73 m^2^, panel (**B**)). Bars: mean change ± SEM. * *p* < 0.0001 vs. baseline, ** *p* non-significant vs. baseline, *** *p* = 0.02 vs. baseline.

**Table 1 jcm-11-05622-t001:** Baseline characteristics. Data: percentage or mean (SD), except * median (IQR). BMI: body mass index. eGFR: estimated glomerular filtration rate. UACR: urine albumin-to-creatinine ratio. ACEis: angiotensin-converting enzyme inhibitors. ARBs: angiotensin II receptor blockers.

Baseline Characteristics	Value
Number of patients	583
Follow-up time (months) *	13.0 (6.4–24.8)
Gender (male/female)	55.4/44.6
Age (years)	60.4 (11.3)
Duration of T2DM (years) *	11.2 (6.4–16.5)
HbA1c (%)	7.76 (1.36)
Weight (kg)	93.4 (21.1)
BMI (kg/m^2^)	34.7 (7.3)
eGFR (ml/min/1.73 m^2^)	85.6 (16.7)
UACR (mg/g Cr) *	7.6 (2.4–26.9)
Chronic kidney disease (%)	
• Stage G0/G1	49.5
• Stage G2	42.0
• Stage G3a	8.0
• Stage G3b	0.4
• Stage G4	0.2
• Stage G5	0
• Stage A1	77.2
• Stage A2	18.8
• Stage A3	4.0
Hypertension	78.7
Hypercholesterolemia	85.6
Hypertriglyceridemia	46.0
Combined hyperlipidemia	42.5
Current smoker	13.8
Diabetic retinopathy	16.0
Diabetic neuropathy	9.8
Coronary artery disease	9.6
Stroke	3.8
Peripheral artery disease	6.3
Arrhythmias	4.6
Heart failure	2.4
ACEis (%)	31.9
ARBs (%)	37.6
Thiazides (%)	31.4
Loop diuretics (%)	6.2

**Table 2 jcm-11-05622-t002:** Predictive factors of eGFR decline after canagliflozin therapy. Data: mean difference (95% CI). UACR: urine albumin-to-creatinine ratio. ARBs: angiotensin II receptor blockers. * *p* < 0.05.

Baseline Variable	Mean Change (95% CI) (Unadjusted)	Mean Change (95% CI) (Adjusted)	Mean Change (95% CI) (Best Model)
Age (per year)	−0.08 (−0.16; −0.003) *	−0.14 (−0.24; −0.04) *	−0.15 (−0.24; −0.05) *
Hypertension (yes vs. no)	−3.34 (−5.44; −1.24) *	−1.91 (−4.74; 0.92)	−2.00 (−4.48; 0.40)
Combined hyperlipidemia (yes vs. no)	−2.52 (−4.27; −0.76) *	−2.59 (−4.66; −0.51) *	−3.09 (−5.00; −1.17) *
Heart failure (yes vs. no)	−6.11 (−11.55; −0.68) *	−5.85 (−11.04; −0.66) *	−5.40 (−10.42; −0.37) *
eGFR (per mL/min)	−0.14 (−0.20; −0.09) *	−0.20 (−0.27; −0.13) *	−0.20 (−0.26; −0.13) *
Proteinuria (vs. UACR < 30 mg/g)	−10.70 (−16.28; −5.13) *	−11.12 (−17.00; −5.25) *	−11.04 (−16.22; −5.85) *
Hematocrit (per 1%)	0.22 (0.03; 0.40) *	0.02 (−0.21; 0.24)	
ARBs (yes vs. no)	−2.36 (−4.17; −0.56) *	−0.94 (−3.15; 1.28)	

## Data Availability

The data are not publicly available due to privacy restrictions.

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
