# Peer review of "Predictive Factors of Renal Function Decline in Patients with Type 2 Diabetes Treated with Canagliflozin in the Real-Wecan Study"

_jcm, 2022, doi:10.3390/jcm11195622_

Round 1
Reviewer 1 Report (Previous Reviewer 1)
the changes made by the authors have significantly improved the paper which will contribute to increasing the knowledge on the use of sglt2i in clinical practice made by the authors have improved the paper.
Author Response
Many thanks for your kind comment and your prior suggestions.
Reviewer 2 Report (New Reviewer)
A very well written manuscript with the aim of answering the precisely framed question on the long-term effect on renal function following canagliflozin therapy.
However, the authors have not evaluated the other SGLT-2 inhibitors such as dapagliflozin and empagliflozin on renal function.
A note on the possible mechanism of action of canagliflozin may be added on the effect on renal function.
Author Response
Reviewer 2.
- A very well written manuscript with the aim of answering the precisely framed question on the long-term effect on renal function following canagliflozin therapy.
- Response: Many thanks for your kind comment
- However, the authors have not evaluated the other SGLT-2 inhibitors such as dapagliflozin and empagliflozin on renal function.
- Response: The REAL WECAN study focused on patients who started canagliflozin 100 mg and patients with prior SGLT-2i treatment who switched to canagliflozin 300 mg. Unfortunately, eGFR trajectories with other SGLT-2is prior to switching to canagliflozin 300 mg were not collected, so renal data with empagliflozin or dapagliflozin cannot be shown in this study. This limitation has been included in the discussion (see limitations, paragraph in red).
- A note on the possible mechanism of action of canagliflozin may be added on the effect on renal function.
- Response: We already discussed the potential nephroprotective mechanisms of SGLT-2is in the introduction. (“Proposed mechanisms responsible for renal protection with SGLT-2is are probably multifactorial and include reductions in blood pressure (BP), body weight and HbA1c, decrease in intraglomerular pressure due to tubule-glomerular feedback activation, reduction in albuminuria, improved cardiac function with maintenance of renal perfusion, activation of anti-inflammatory and antifibrotic pathways and better oxygenation of tubular cells in the proximal tubule”). Notwithstanding, we have included a new paragraph in the discussion and a new reference.
“As we pointed out in the introduction, several mechanisms may contribute to the nephroprotective effects seen with canagliflozin. Well known metabolic effects include reduction of glucotoxicity and perivisceral fat. Moreover, and increase in beta-hydroxybutyrate levels, a substrate with high energy efficiency for the distal nephron, may protect the renal medulla from hypoxia. Canagliflozin has systemic hemodynamic effects, as it lowers the systolic BP transmitted to the kidney, improves heart function and reduces the need of loop diuretics, which avoids intravascular volume depletion. Intrarenal hemodynamic effects are probably the most important renoprotective mechanism of canagliflozin. Restoration of tubule-glomerular balance, providing more sodium to the macula densa, normalizes the tone of the afferent arteriole and is followed by long-term eGFR stabilization and reduction of albuminuria. Finally, reduction of glucotoxicity and proteinuria and the activation of anti-inflammatory and antifibrotic pathways by canagliflozin may play a joint role in reducing tubulointerstitial damage in the diabetic kidney.”
Reference 20: Weir MR, McCullough PA, Buse JB, Anderson J. Renal and Cardiovascular Effects of Sodium Glucose Co-Transporter 2 Inhibitors in Patients with Type 2 Diabetes and Chronic Kidney Disease: Perspectives on the Canagliflozin and Renal Events in Diabetes with Established Nephropathy Clinical Evaluation Trial Results. Am J Nephrol. 2020;51(4):276-288.
This manuscript is a resubmission of an earlier submission. The following is a list of the peer review reports and author responses from that submission.
Round 1
Reviewer 1 Report
The manuscript of Gorgojo-Martinez et al. focuses on factors that can predict eGFR reduction in patients with type 2 diabetes mellitus treated with canaglifozin in a real world setting.
Despite some clinical trials have shown that long-term treatment with SGLT2i is protective against renal function, in some cases there is a worsening of eGFR so it is important to identify the patients who more easily undergo a restriction of the filtrate, to improve their clinical surveillance.
In addition, renal complication is very common among patients with type 2 diabetes mellitus and plays a causal role for CV pathologies.
Did the cohort of patients taking canaglifozin 300 mg day also contain those patients who were included in the canaglifozin 100 mg day patient group?
If so, could you explain the reason for this choice in the text?
Is it possible to sub-analyze patients with longer follow-up to check the EGFR trend in these patients?
Reviewer 2 Report
I have read the paper with interest, expecting however results regarding acute GFR drop after the start of the treatment, especially as the author refer to the post-hoc CREDENCE analysis published by Oshima et al in 2021.
This is in my opinion the main flaw - there is no chance to capture anything acute because the authors have the second visit 6+/-2 months after the first one. At the six months they capture the slope already increasing after the initial drop, not knowing how low it was. Additionally, having such a large time span makes this time point absolutely unreliable, and what the author show on graphs as "6 months" may be also 7 or 5. The time span should be distinctly shown on the graph (6+/-2 months). Also, what is exactly "last visit" on the time axis? What was the time span in which the last visits have taken place? Is it justified to show that on the time axis as one time-point? Surely not. How to show that on the graph not suggesting that the time between visits was similar (as they are suggesting now) - I do not know.
Again - what was the time of observation? And what was an endpoint? Was there any? In the methodology the authors write about "estimating of the GFR slope", but what endpoint is it?
There is unfortunately no control group not treated with flozins. Therefore we do not know, whether the decline shown in the graphs is lower (less steep) or not. It is also impossible to include flozin use or dose into multivariate analysis, which would be interesting (especially dose).
The next flaw is that the authors combine patients who just started the therapy with others, who were already treated. The time of treatment is not given. These two cohorts are in my opinion incomparable, because of the initial GFR drop in newly treated patients, because possibly long traetment in the other group, and must not be assessed as a combined population. From the other side, stratifying the group for two small groups in a RWE study makes not much sense taking into account the reliability.
Only few patients in this study had a meaningful (>30%) drop of the GFR. This group is to small to draw any reliable conclusion, it is anyway not characterized. Was this group clinically diagnosed? Was the reason for a rapid decline searched for? Had they kidney biopsies?
The authors do not compare the basal parameters of the groups. Even if compared, the proper methodology in RWE studies when one builds two groups is a propensity matching, which was not done here.
The methodolgy of the "multivariate analysis" and the message of Table 2 is unclear, and the Table is not self-explaining. How was the "decline" defined? What is "greater decline"? The difference shown int he Table is between what and what? What are the odds ratios and why are they not shown? Why has ARB use apparently similar influence on GFR as heart failure (negative difference in both cases)?
The conclusion in abstract are totaly different from those in the main text
